# Effect of *HFE* Gene Mutations on Iron Metabolism of Beta-Thalassemia Carriers

**María E. Mónaco [1], Natalia S. Alvarez Asensio [2], Cecilia Haro [2], Magdalena M. Terán [2],
Miryam E. Ledesma Achem [2], Blanca A. Issé [2] and Sandra S. Lazarte [2,\***

[1] Instituto de Biología, Facultad de Bioquímica, Química y Farmacia, Universidad Nacional de Tucumán, Chacabuco 461, San Miguel de Tucumán CP4000, Argentina

[2] Instituto de Bioquímica Aplicada, Facultad de Bioquímica, Química y Farmacia, Universidad Nacional de Tucumán, Balcarce 747, San Miguel de Tucumán CP4000, Argentina

\* Correspondence: sandra.lazarte@fbqf.unt.edu.ar; Tel.: +54-9-381-6648653

**Abstract:** The human hemochromatosis protein HFE is encoded by the *HFE* gene and participates in iron regulation. The aim of this study was to detect the most frequent *HFE* gene mutations in a control population and in β-thalassemia trait (BTT) carriers, and to study their relationship with iron metabolism. Total blood count, hemoglobin electrophoresis at alkaline pH, HbA$_2$ quantification, iron (Fe), total Fe binding capacity and ferritin were assayed. *HFE* gene mutations were analyzed by real-time PCR. A total of 119 individuals (69 normal and 50 BTT) were examined. In the control group, 9% (6/69) presented a codon 282 heterozygous mutation (C282Y), and 19% a codon 63 mutation (H63D) (13/69, 11 heterozygotes and 2 homozygotes). In the BTT group, 3 carriers (6%) were heterozygous for C282Y, 14 (28%) for H63D, 1 (2%) for a codon 65 mutation and 1 (2%) was H63D and C282Y double heterozygous. Control group Fe metabolism did not show significant differences ($p > 0.05$) according to whether or not they carried an *HFE* gene mutation; while the BTT group with and without *HFE* mutation showed higher Fe and ferritin than the control group ($p < 0.05$). However, no increases in iron parameters were detected in BTT carriers that simultaneously exhibited an H63D mutation compared to BTT subjects without a mutation. Therefore, the iron metabolism alterations observed in BTT carriers could not be attributed to the presence of *HFE* gene mutations. It is likely that BTT individuals have other genetic modifiers that affect their iron balance.

**Keywords:** HFE; beta-thalassemia; hereditary hemochromatosis; iron metabolism





## 1. Introduction

Iron is an essential trace element required as a component of molecules sensing, transporting, and storing oxygen; as well as enzymes involved in oxidation and reduction of substrates during energy production, intermediate metabolism, and the generation of reactive oxygen or nitrogen species for host defense. The impressive advances in our knowledge of iron metabolism have translational implications for the diagnosis and treatment of iron disorders. This large group of pathological condition includes both genetic and acquired disorders, which can be classified as iron deficiency (absolute or functional) and iron overload (primary and secondary). Iron overload may occur in genetic disorders and may be primary, such as the different types of haemochromatosis; or be secondary to increased iron absorption accompanied by hereditary anemia with ineffective erythropoiesis, such as beta-thalassemia [1].

Thalassemia is a hereditary anemia resulting from defects in hemoglobin (Hb) production. Hemolysis and ineffective erythropoiesis together cause the anemia that occurs in thalassemia. Beta (β)-thalassemia, which is caused by a decrease in the production of β-globin chains, affects multiple organs and is associated with considerable morbidity and mortality. More than 300 β-thalassemia alleles have now been characterized, but only about 40 account for 90% or more of all β-thalassemia cases worldwide. The vast

majority of β-thalassemia is caused by point mutations and, rarely, by deletions. Patients with β-thalassemia have been typically categorized as minor, major, or intermedia on the basis of α-globin or β-globin chain imbalance, severity of anemia, and clinical picture at presentation. β-thalassemia minor (trait or carrier) represents the heterozygous inheritance of a β-thalassemia mutation, with patients often having clinically asymptomatic microcytic anemia [2].

Tissue iron overload is the most important complication of β-thalassemia and is a major focus of therapeutic management. Blood transfusion is a comprehensive source of iron loading for β-thalassemia patients. Nevertheless, iron overload also occurs in patients who have not received transfusions such as patients suffering from thalassemia intermedia [3]. Hepcidin concentrations are suppressed in β-thalassemia major, intermedia and minor [4–6]. This fact allows excessive iron absorption and development of systemic iron overload [4].

Hereditary hemochromatosis (HH) is the most common genetic iron overload disorder among Caucasians. Although multiple mutations can lead to the clinical syndrome, the most common mutations are those in the *HFE* gene [7]. In 1996, Feder and colleagues used positional cloning to identify *HFE*, which is linked to the major histocompatibility complex (MHC) on chromosome 6p [8].

The HFE protein is required for normal regulation of hepatic synthesis of hepcidin, the main controller of iron metabolism [1]. Common *HFE* mutations account for almost 90% of hemochromatosis phenotypes in whites of Western European descent [8].

The most prevalent disease-causing *HFE* mutation in the general population is the 845G polymorphism, which causes a p.Cys282Tyr amino acid substitution (C282Y) in the HFE protein. Homozygosity for this single missense substitution (p.Cys282Tyr) is responsible for between 60 and 100% of cases of HH among European populations [8]. *HFE* gene sequencing approaches have identified additional *HFE* mutations with different pathological impacts. These include the amino acid alterations H63D and S65C [9].

Although the causes of iron accumulation in subjects with β-thalassemia are clear, the fact that this accumulation occurs in greater proportions in some patients suggests a possible association between HH and β-thalassemia. The presence of *HFE* gene mutations in patients affected with β-thalassemia leads to the expression of phenotypic hematological characteristics different from those that would be observed with only primary mutations of the β-globin gene.

The objective of this study was to compare the results of blood testing in a group of subjects with β-thalassemia trait (BTT) and normal individuals with and without common *HFE* mutations, and to investigate the relationship between *HFE* mutations and iron metabolism.

## 2. Materials and Methods

### 2.1. Subjects

A prospective cross-sectional analytical study was carried out. The population was composed of apparently healthy individuals and patients with BTT, who were diagnosed in the Laboratory of Hematology of the Instituto de Bioquímica Aplicada of the Universidad Nacional de Tucumán (UNT), during the period 2016–2018. All patients and controls signed an informed consent form. The study protocol and the declaration of informed consent were approved by the Comité de Bioética de la Facultad de Medicina of UNT.

### 2.2. Hematological and Molecular Analysis

A blood sample was collected with EDTA-K$_2$, and a fraction without anticoagulant for serum determinations. Blood count was performed using a Sysmex KX-21N hematological counter (Kobe, Japan). β-thalassemia diagnosis was made by Hb electrophoresis at alkaline pH, and HbA$_2$ quantification through microcolumn chromatography (ByoSystems, Barcelona, Spain). Iron metabolism was assayed by quantifying serum iron (Fe), and total iron binding capacity (TIBC) by a colorimetric method (Wiener lab, Rosario,

Argentina). Ferritin was measured by electrochemiluminescence (COBAS, Roche, Basel, Switzerland). Transferrin saturation (SAT) was calculated with the following formula: SAT [%] = (Fe/TIBC) × 100

Isolation of genomic DNA was carried out with the High Pure PCR Template Preparation Kit (Roche Diagnostics) from 200 μL of whole blood anticoagulated with EDTA-K$_2$. Characterization of mutations in the β-globin gene and the *HFE* gene was carried out using real-time polymerase chain reaction (PCR) with FRET probes (fluorescence resonance energy transfer). PCR, melt curves, and subsequent analysis were performed on the Light-Cycler 2.0 equipment (Roche). A 587 bp region of the β-globin gene was amplified for β-thalassemia mutations [10]:

−  Forward Primer: 5′-GCTGTC ATCACTTAGACCTCA-3′.
−  Reverse Primer: 5′-CACAGTGCAGCTCA CTCAG-3′.

Two combinations of hybridization probes were used according to the following scheme [11]: SET A: Mutations: IVSI-110 G → A; Cd39 C → T; SET B: Mutations: IVSI-1 G → A; IVSI-5 G → A; IVSI-6 T → C.

*HFE* gene mutations were detected with the following pairs of primers and probes [12]:

C282Y mutation, forward primer: CTGGATAACCTTGGCTGTACC, and reverse primer: GGCTCTCATCAGTCACATACC.

H63D/S65C mutations, forward primer: GTCTCCAGGTTCACACTCTC and reverse primer: CCATAATAGTCCAGAAGTCAACAG.

### 2.3. Statistical Analysis

Statistical analysis was carried out using SPSS 20.0. The results were reported as median and range. Mann–Whitney U tests were used for the comparisons and a significance level of $p < 0.05$ was adopted.

## 3. Results

In total, 69 apparently healthy subjects (48 women and 21 men) aged 21–64 years, and 50 BTT carriers (31 women and 19 men) aged 16–79 years were studied. Table 1 shows the *HFE* gene mutations detected in the population. *HFE* mutations were present in 28% (19/69, 95% CI = 18–39%) of the control subjects, and in 38% (19/50, 95% CI = 28–54%) of the BTT group (Table 1). No significant difference was observed between control and BTT frequency of *HFE* mutation ($p > 0.05$).

**Table 1.** Mutations in the *HFE* gene in β-thalassemia trait and apparently healthy subjects.

| Genotype | Control (*n* = 69) | BTT (*n* = 50) |
|---|---|---|
| H63D/wt | 11 (16%) | 14 (28%) |
| H63D/H63D | 2 (3%) | 0 |
| C282Y/wt | 6 (9%) | 3 (6%) |
| S65C/wt | 0 | 1 (2%) |
| H63D/C282Y | 0 | 1 (2%) |
| wt/wt | 50 (72%) | 30 (60%) |

Abbreviations: wt, wild type.

Table 2 shows the relationship between ethnicity and β-thalassemia and HFE mutations identified in BTT carriers. Italian origin predominated in the BTT population, followed by Spanish and Arabic. The three origins represented almost 90% of the total population. The rest were of Creole descent, and from France, Bulgaria and Germany. Most of the *HFE* mutations were detected in Italian-descent individuals.

**Table 2.** Relationship between ethnic origin and mutations detected in heterozygous β-thalassemia subjects.

| Ethnicity | Beta-thalassemia Mutation | | | | | | Total | HFE Mutation | | | |
|---|---|---|---|---|---|---|---|---|---|---|---|
| | CD39 | IVSI-1 | IVSI-110 | IVSII-1 | IVSII-745 | ND | | C282Y | H63D | S65C | WT |
| Spanish | 1 | 3 | 1 | 0 | 0 | 2 | 7 | 0 | 0 | 0 | 7 |
| Italian | 6 | 1 | 5 | 1 | 0 | 1 | 14 | 1 * | 5 | 1 | 7 |
| Arab | 0 | 4 | 0 | 0 | 0 | 1 | 5 | 0 | 2 | 0 | 3 |
| Spanish–Italian | 2 | 0 | 2 | 0 | 1 | 1 | 6 | 1 | 2 | 0 | 3 |
| Spanish–Arab | 0 | 3 | 1 | 0 | 0 | 1 | 5 | 1 | 1 | 0 | 3 |
| Creole | 1 | 2 | 1 | 0 | 0 | 0 | 4 | 0 | 1 | 0 | 3 |
| Arab–French | 0 | 1 | 0 | 0 | 0 | 0 | 1 | 0 | 0 | 0 | 1 |
| Spanish–Italian–Arab | 0 | 0 | 2 | 0 | 0 | 0 | 2 | 0 | 1 | 0 | 1 |
| Spanish–Italian–Arab–German | 0 | 1 | 0 | 0 | 0 | 0 | 1 | 1 | 0 | 0 | 0 |
| Spanish–Italian–Bulgarian | 0 | 0 | 1 | 0 | 0 | 0 | 1 | 0 | 1 | 0 | 0 |
| French | 0 | 0 | 0 | 0 | 0 | 1 | 1 | 0 | 0 | 0 | 1 |
| Spanish–Arab–Bulgarian | 0 | 0 | 0 | 0 | 0 | 1 | 1 | 0 | 0 | 0 | 1 |
| Unknown | 0 | 1 | 0 | 0 | 0 | 0 | 2 | 0 | 2 | 0 | 0 |
| Total | 10 | 16 | 14 | 1 | 1 | 8 | 50 | 4 | 15 | 1 | 30 |

Abbreviations: ND, not determined; WT, wild type. * Double heterozygous C282Y/H63D.

The most frequent β-thalassemia mutation was IVSI-1 (32%, 16/50), followed by IVSI-110 (28%, 14/50), CD39 (20%, 10/50), IVSII-745 (2%, 1/50), and IVSII-1 (2%, 1/50). In eight (16%) BTT subjects, the mutation could not be determined. There were no significant differences ($p > 0.05$) in the iron parameters according to β-thalassemia mutation type ($\beta^0$ or $\beta^+$), except for TIBC which was higher in the $\beta^+$ group with *HFE* gene mutation (Table 3).

**Table 3.** Influence of the presence of *HFE* gene mutation on iron metabolism according β-thalassemia mutation type.

| β-thalassemia Mutation Type | HFE Gene Mutation | Fe (µg/dL) | TIBC (µg/dL) | Transferrin Saturation (%) | Ferritin (ng/mL) |
|---|---|---|---|---|---|
| $\beta^0$ (IVSI-1, CD39, IVSII-1) $n = 27$ | Present, $n = 9$ | 112 (72–139) | 286 (262–325) | 39 (25–48) | 203.0 (41.6–626.0) |
| | Absent, $n = 18$ | 102 (73–165) | 287 (223–392) | 33 (26–56) | 196.7 (32.6–833.0) |
| $\beta^+$ (IVSI-110, IVSII-745) $n = 15$ | Present, $n = 8$ | 120 (52–160) | 311 (264–416) * | 42 (12–51) | 203.7 (45.1–884.7) |
| | Absent, $n = 7$ | 98 (63–156) | 299 (276–321) | 33 (21–48) | 254.2 (18.6–359.0) |
| Not determined $n = 8$ | Present, $n = 2$ | 105 (105–105) | 331 (302–359) | 32 (29–35) | 148.8 (45.3–252.4) |
| | Absent, $n = 6$ | 98 (58–154) | 299 (259–393) | 32 (20–60) | 258.4 (55.7–1478.0) |

Abbreviations: TIBC, total iron binding capacity. * $p < 0.05$ compared to the $\beta^0$ group with *HFE* gene mutation.

In the BTT group, it was observed that transferrin saturation was higher in subjects of Italian and Arab origin compared to those of Spanish origin ($p < 0.05$) which did not have mutations in the *HFE* gene (Table 2). In contrast, serum iron, TIBC and ferritin levels did not show significant changes ($p > 0.05$).

Table 4 displays the hematological study results. As expected, significant differences were detected ($p < 0.05$) in red blood parameters between BTT male and female groups compared to controls without BTT. Hematocrit and MCHC showed no significant differences ($p > 0.05$) between the male groups. In those groups without *HFE* mutation, serum iron saturation and ferritin were higher in the control male group and the BTT female group than the control female group ($p < 0.05$). In those groups with *HFE* mutation, only ferritin showed a significant difference in the BTT female group compared to the control female group ($p < 0.05$). Iron parameters did not differ in BTT carriers based the presence or absence of *HFE* mutation ($p > 0.05$).

**Table 4.** Hematological parameters according to *HFE* gene mutation presence in the study groups.

| Parameter | Control without *HFE* Gene Mutation | | Control with *HFE* Gene Mutation | |
|---|---|---|---|---|
| | Female (*n* = 32) | Male (*n* = 18) | Female (*n* = 16) | Male (*n* = 3) |
| Age (years) | 28 (22–64) | 31 (21–63) | 26 (24–37) | 39 (21–42) |
| RBC (×$10^{12}$/L) | 4.42 (3.97–5.02) | 4.95 (4.37–5.43) [a] | 4.38 (3.84–5.03) | 4.96 (4.36–5.65) |
| Ht (L/L) | 0.40 (0.37–0.44) | 0.44 (0.41–0.50) [a] | 0.41 (0.35–0.44) | 0.44 (0.39–0.47) |
| Hb (g/L) | 126 (114–141) | 146 (133–164) [a] | 130 (113–137) | 148 (127–161) |
| MCV (fL) | 90.0 (83.0–95.0) | 89.5 (83.2–96.0) | 91.4 (82.0–98.3) [a] | 88.0 (82.8–89.7) |
| MCH (pg) | 29 (26–31.2) | 30 (28–33) | 29.4 (25.0–31.2) | 29.0 (28.5–29.8) |
| MCHC (g/L) | 321 (308–338) | 329 (317–372) | 318 (298–338) | 333 (326–344) |
| RDW (%) | 12.8 (11.2–17.4) | 12.7 (12.0–13.7) | 12.7 (11.5–13.5) | 12.4 (12.3–12.9) |
| Iron (µg/dL) | 76.5 (46–153) | 98.5 (57–167) [a] | 94 (53–159) | 72 (64–135) |
| TIBC (µg/dL) | 314 (200–462) | 284 (209–372) | 288 (213–400) | 264 (263–296) |
| Sat (%) | 24 (15–53) [b] | 32 (21–50) [a] | 32 (19–47) | 27 (22–51) |
| Ferritin (ng/mL) | 57.6 (13.2–959.9) | 200.6 (47–930.5)[a] | 52.8 (26.3–147) | 514.9 (83.8–574.7) |
| | **BTT without *HFE* gene mutation** | | **BTT with *HFE* gene mutation** | |
| | Female (*n* = 19) | Male (*n* = 12) | Female (*n* = 12) | Male (*n* = 7) |
| Age (years) | 38 (21–79) | 36 (16–75) | 35 (16–69) | 52 (29–65) |
| RBC (×$10^{12}$/L) | 5.52 (4.40–6.10) [a,b] | 6.12 (5.06–7.14) [c,d] | 5.62 (4.89–7.39) [a,b] | 6.32 (5.46–7.10) [c,d] |
| Ht (L/L) | 0.35 (0.30–0.40) [a,b] | 0.41 (0.35–0.46) | 0.36 (0.33–0.47) [a,b] | 0.41 (0.35–0.46) |
| Hb (g/L) | 107 (89–123) [a,b] | 122 (105–137) [c,d] | 111 (99–135) [a,b] | 121 (103–137) [c,d] |
| MCV (fL) | 65.3 (62.1–72.7) [a,b] | 66.4 (60.8–69.5) [c,d] | 64.9 (61.0–74.0) [a,b] | 64.1 (60.8–69.6) [c,d] |
| MCH (pg) | 19.5 (17.9–22.3) [a,b] | 198 (17.9–21.8) [c,d] | 19.6 (18.1–22.7) [a,b] | 19.0 (17.6–21.6) [c,d] |
| MCHC (g/L) | 296 (216–316) [a,b] | 301 (294–317) | 302 (205–317) [a,b] | 296 (289–310) |
| RDW (%) | 15.8 (13.2–18.4) [a,b] | 16.8 (14.4–19.9) [c,d] | 16.1 (14.2–17.0) [a,b] | 16.4 (15.7–18.0) [c,d] |
| Iron (µg/dL) | 98 (63–131) [a] | 104.5 (58–165) | 114 (52–152) | 102 (68–160) |
| TIBC (µg/dL) | 280 (223–393) | 298 (259–353) | 301 (264–416) | 307 (262–360) |
| Sat (%) | 32 (21–56) [a] | 34 (20–60) | 40 (12–51) | 39 (22–48) |
| Ferritin (ng/mL) | 165.6 (18.6–833) [a] | 298.2 (97.7–1478) | 137.8 (41.6–308) [b,e] | 510 (163–884.7) |

Abbreviations: RBC, red blood count; Ht, hematocrit; Hb, hemoglobin; MCV, mean corpuscular volume; MCH, mean corpuscular hemoglobin; MCHC, mean corpuscular hemoglobin concentration; RDW, red blood cell distribution width. [a] $p < 0.05$ compared to control females without *HFE* gene mutation; [b] $p < 0.05$ compared to control females with *HFE* gene mutation; [c] $p < 0.05$ compared to control males without *HFE* gene mutation; [d] $p < 0.05$ compared to control males with *HFE* gene mutation; [e] $p < 0.05$ compared to BTT males with *HFE* gene mutation.

Figure 1 shows levels of serum iron (Fe), TIBC, transferrin saturation (TS) and ferritin in control and BTT groups according *HFE* mutation type. Iron metabolism analysis in the apparently healthy group did not reveal significant differences ($p > 0.05$) according to presence or absence of *HFE* gene mutation. The BTT group demonstrated higher Fe, ST and ferritin than controls ($p < 0.05$). The BTT group with H63D mutation only exhibited an increase in ferritin compared to controls with the same mutation ($p < 0.05$). Only one male BTT carrier without an *HFE* gene mutation showed ferritin higher than 1000 ng/mL. The only BTT patient with the C282Y/H63D genotype and the two control H63D homozygotes were females and did not show alterations in iron metabolism. Comparisons according to the type of mutation (C282Y or H63D) with respect to sex could not be made due to insufficient numbers of subjects in each group.

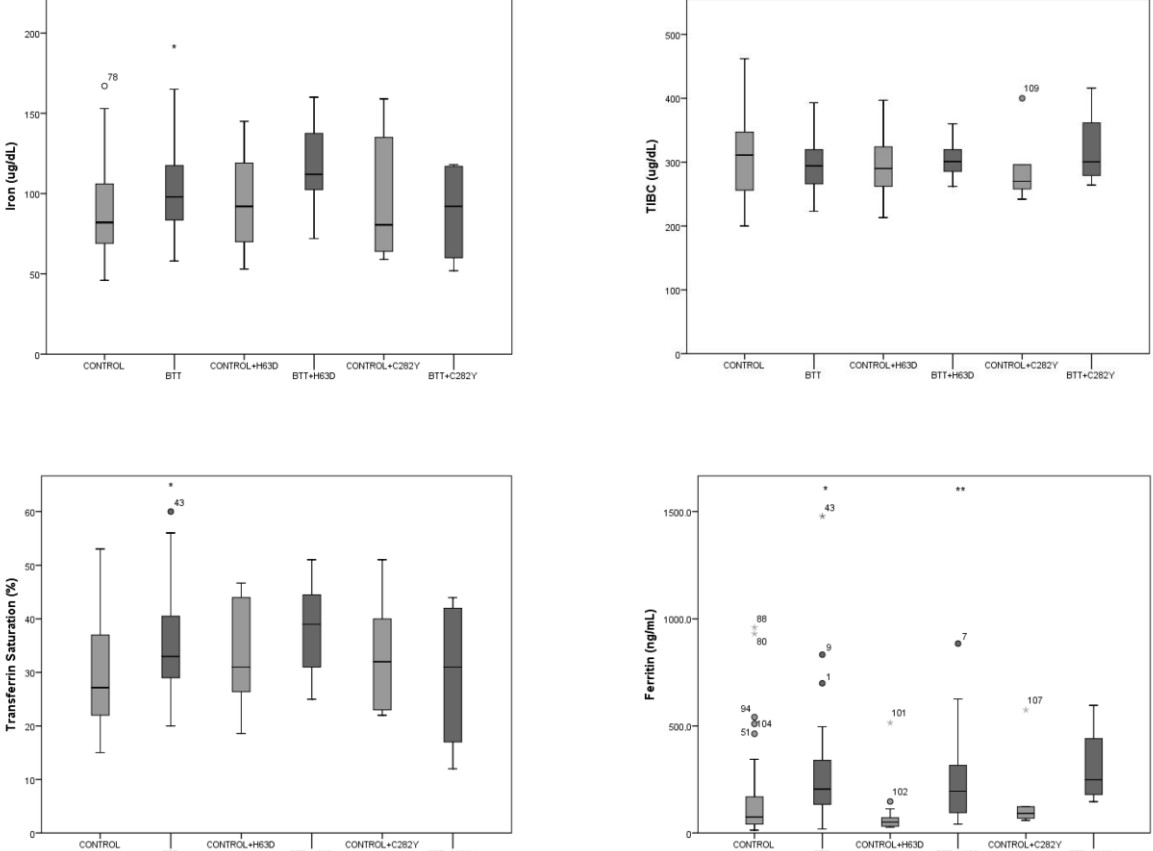

**Figure 1.** Iron parameters according the presence of *HFE* mutation in the groups under study. Abbreviations: TIBC, total iron binding capacity. * *p* < 0.05 compared to control; ** *p* < 0.05 compared control + H63D. The number case is included in outliers.

## 4. Discussion

β-thalassemia trait is characterized by ineffective erythropoiesis, which can induce excess iron absorption and cause iron overload. The interaction of β-thalassemia with hemochromatosis, which is caused in most cases by mutations in the *HFE* gene, can further exacerbate iron overload. Previous work has described an increase in iron, transferrin and/or ferritin levels in β thalassemia carriers that also had an H63D mutation, compared to those that did not carry this mutation [13–16]. However, other publications showed that the simultaneous presence of β-thalassemia and an *HFE* gene mutation did not modify serum ferritin levels [17–20].

Iron metabolism analysis revealed that β-thalassemia carriers had higher Fe, ST and ferritin than apparently healthy subjects. When comparing β-thalassemia subjects with an H63D mutation to controls with same mutation, an increase in ferritin level was observed in the first group. Recent publications have found an increase in ferritin levels in β-thalassemia major individuals with an H63D mutation [21,22]. In present work, this rise could not be attributed to the presence of the *HFE* gene mutation, since it was shown that BTT carriers displayed higher ferritin levels than healthy subjects. Another study previously described this effect in both BTT women and men [23,24]. Increased iron storage in BTT, as reflected by significantly higher concentrations of ferritin, would be a consequence of higher iron absorption [23].

As with other studies, our comparison between $β^0$ and $β^+$ carriers with or without *HFE* gene mutation did not detect significant differences in parameters related to iron, except TIBC, which was significantly increased in the $β^+$ group with *HFE* mutation [19,20].

On the other hand, the presence of *HFE* gene mutation in the control population did not increase iron parameters. This fact has already been observed by other authors [25,26]. However, there are authors that affirm the opposite [27,28]. Jackson et al. detected significant differences in serum iron, TIBC and transferrin saturation between blood donors with and without *HFE* mutation; but H63D heterozygotes and C282Y heterozygous females did not show significant differences in ferritin levels when compared with wild type donors. In the present work, the analysis of iron probably did not exhibit significant differences because most of the participants heterozygous for the H63D mutation were female. Serum iron and ferritin are influenced by age, sex, diet and disease, as well as by biological variation among individuals and by methodological differences [29].

The spectrum of β-thalassemia mutations was different to that previously reported for Tucumán [11]. In our study, the IVSI-110 mutation was the most frequent, whereas the CD39 mutation was most prevalent in the first molecular study conducted in the region. IVSI-110 was present mainly in Italian-origin carriers. The frequency of IVSI-110 in Italy varies among the different geographical regions, appearing in some places with a greater or similar frequency as CD39 [30,31]. Additionally, the Arab contribution to IVSI-110's origin in the Tucumán population cannot be discounted, since it is the most prevalent in Syria and Lebanon [32,33]. In addition, ethnicity was an important factor in the analysis of HFE genotype/phenotype relationships. Thus, in both Italian- and Arab-origin BTT individuals, it was shown that transferrin saturation was increased compared to Spanish-origin individuals in whom *HFE* mutation was not detected. In a Spanish study, the C282Y heterozygote, H63D heterozygote and homozygote, and H63D/S65C compound heterozygote genotypes were associated with increased transferrin saturation relative to wild type in the general population [26].

Other studies have shown that *HFE* gene mutations are common among β-thalassemia carriers compared with normal controls [34,35]. This was not confirmed in present work since there was no significant increase in *HFE* mutation frequency in the BTT group. The presence of the C282Y/H63D genotype was related to iron overload [36], but the only BTT patient with this genotype did not display iron elevation. Jaing et al. found that β-thalassemia carriers who are homozygous for the H63D variant have higher serum ferritin levels than carriers without the variant [37]. However, in the present study there was no iron metabolism disturbance in H63D homozygotes. This effect could be a consequence of the low penetrance of this genotype, and that other cofactors that contribute to iron overload would be usually present, such as iron intake, alcohol intake, tobacco use, and male gender [7,26].

A limitation of our study is that other mutations that could increase iron deposits were not analyzed in current work, such as mutations in hemojuvelin, hepcidin, transferrin receptor 2 and ferroportin genes [38]. Additionally, acquired factors such as dietary habits, blood donations, pregnancies, menopause, and malabsorption syndromes, among others, could influence the iron metabolism results [20].

## 5. Conclusions

There is a wide spectrum of phenotypes in β-thalassemia, and there are population-specific genetic modifiers that influence them. These include mutations in the *HFE* gene [37]. In BTT subjects of Italian and Arab origin, only a significant increase in transferrin saturation was detected, compared to subjects of Spanish origin. Remarkably, in the latter, mutations in the *HFE* gene were not identified. On the other hand, the present work detected no increase in iron parameters in β-thalassemia carriers that simultaneously exhibited *HFE* mutation compared to those BTT subjects without *HFE* mutation. In addition, apparently healthy subjects with *HFE* mutation did not show differences in iron status. However, BTT individuals displayed higher ferritin levels than controls in both groups, with and without *HFE* mutation. Because of this, there may be other genetic modifiers present in β-thalassemia individuals that could aggravate their ferric balance.

**Author Contributions:** Conceptualization and methodology, M.E.M., B.A.I. and S.S.L.; formal analysis (sample collection, DNA extraction, beta-thalassemia diagnosis and *HFE* genotyping), N.S.A.A., C.H., M.M.T. and M.E.L.A.; data curation, M.E.M., N.S.A.A. and S.S.L.; writing—original draft preparation, M.E.M. and S.S.L.; writing—review and editing, S.S.L. and B.A.I.; supervision, project administration and funding acquisition, M.E.M., B.A.I. and S.S.L. All authors have read and agreed to the published version of the manuscript.

**Funding:** This research was funded by the Consejo de Investigaciones de la Universidad Nacional de Tucumán (CIUNT 26/D520) and Alberto J. Roemmers Foundation.

**Institutional Review Board Statement:** The study was conducted in accordance with the Declaration of Helsinki, and approved by the Comité de Bioética de la Facultad de Medicina of Universidad Nacional de Tucumán, Argentina.

**Informed Consent Statement:** Informed consent was obtained from all subjects involved in the study.

**Data Availability Statement:** The data that support the findings of this study are available from the corresponding author, S.S.L., upon reasonable request.

**Acknowledgments:** The authors thank biochemist specialist Guillermo Vechetti and Laboratorio Tucumán for the use of its molecular biology equipment.

**Conflicts of Interest:** The authors declare no conflict of interest.

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
