# Peer review of "Effect of HFE Gene Mutations on Iron Metabolism of Beta-Thalassemia Carriers"

_thalassrep, doi:10.3390/thalassrep13010010_

Round 1
Reviewer 1 Report
Monaco and colleagues analysed 50 BTT patients with and without HFE mutations comparing iron paramenters with a small cohort of healthy controls. They did not find any differences between two groups suggesting any effects of HFE variant concluding that other factors such acquired or environmental could affect iron metabolism.
The work was well written, with no major criticisms.
N° of Registration in Ethical Committee should be added to line 90.
Only few suggestions.
In order to support and validate their hypothesis adding more value, could Authors assess serum hepcidin values?
Since most of the HFE mutations were detected in Italian descent individuals as reported in lines 133-134, could this different genetic background have a different weight on BTT patients? Could Authors try to assemble patients in three origins groups (Italian, Spanish and Arabic) and compare iron parameters between these?
Due to different genetic background I’d suggest to underline population origin in conclusion and discussion paragraphs.
Reviewer 2 Report
Some clarifications in the discussion and in figure 1 are recommended, as follows
Figure 1
Figure 1 must be broken down between men and women, since in Table 4 clearly different data emerge in some parameters according to the corresponding gender.
Discussion:
Lines: 195-197
Text: “In present work, this rise could not be attributed to the presence of the HFE gene mutation, since it has been shown that the BTT carrier displayed higher ferritin levels than healthy subjects”
Comment: In order to confirm this assertion the authors should indicate whether or not there were differences between the values of the Fe parameters, particularly ferritin, between the BTT with some HFE gene mutation compared to the BTT without said mutations, to discriminate whether the HFE mutations contribute any extra increase in deposits of Fe with respect to the BTT (separated by gender)
Lines: 229-230
Text: “This effect could be consequence of the low penetrance of this genotype, and that other cofactors that contribute to iron overload would be usually present [7, 26].
Comment: The authors should indicate what the mentioned “other cofactors” are.
